# Sustainability of Egyptian Cities through Utilizing Sewage and Sludge in Softscaping and Biogas Production

Usama Konbr [1],*[ID], Walid Bayoumi [2], Mohamed N. Ali [3] and Ahmed Salah Eldin Shiba [4]

[1] Department of Architecture, Faculty of Engineering, Tanta University, Tanta 31733, Egypt
[2] Department of Regional Planning, Faculty of Urban and Regional Planning (FURP), Cairo University, Giza 12613, Egypt; walid.bayoumi@cu.edu.eg
[3] Department of Environment and Sanitary Engineering, Faculty of Engineering, Beni-Suef University, Beni Suef 62511, Egypt; mohamednabil@eng.bsu.edu.eg
[4] Department of Architecture, Faculty of Engineering, Beni-Suef University, Beni Suef 62511, Egypt; ahmed.salah@eng.bsu.edu.eg
* Correspondence: drusamakonbr@f-eng.tanta.edu.eg

**Abstract:** The National Egyptian Agenda 2030 recently adopted the concepts of sustainable cities, mitigating and adapting to climate change. This study responded to these concepts by treating sewage to reuse it in softscaping and recycling sludge to reduce energy consumption and support heating systems by producing biogas. Of the limitations of this study, it focuses on a compound to propose a model to increase the sustainability of Egyptian cities. This study used many technologies, such as biological treatment processes, activated sludge, trickling filters, and fixed bioreactors. However, Membrane bioreactors (MBRs) have seemed to be the most suitable technology because of their low cost and footprint. Additionally, a pilot laboratory was established to simulate the sewage treatment plant. It consisted of a primary sedimentation tank followed by an MBR tank and a chlorine disinfection tank, where the sludge was fed into a cylindrical anaerobic digester. The amount of sludge collected generated 41.5 mL/day of biogas. The application of this large-scale batch reactor will produce around 38 m3/day of biogas. Applying the findings of this study to the treatment and reuse of domestic sewage and sludge can provide up to 50% of the water needed for the green area of the compound.

**Keywords:** sustainable cities; softscaping; sewage and sludge; membrane bioreactors (MBRs); biogas production; anaerobic digester; heating systems; national egyptian agenda 2030

## 1. Introduction

Sustainable cities, ecosystem degradation, urban health, and resilience are critical issues. Scientists have struggled for decades to establish a new defined trend in the global ecological crisis. This trend involves moving urban facilities toward sustainable development linked to the need to modernize the urban and manufactured environment to achieve environmental, social, and economic consistency [1–3]. It also aims to renew settlement planning to provide more opportunities for all, based on improved use of resources and reduced environmental impacts, to improve the overall quality of life [4].

Concerning human wellbeing, endurance led research to find its way to deal with many issues by adopting the advancement of sustainable solutions [5]. These are varied to care for water, energy, and resources. Recently, there has been a great deal of interest from large-scale investors and public support for new energy sources, and one of the biggest drawbacks is the high investment costs [6]. Today, residual zones use about 40% of the total energy used for various human activities, such as heating. By 2050, this percentage of energy use will increase to 50%. Various sources of energy are used to generate electricity and heat to meet the needs of society, such as fossil fuels, neutral gas, coal, and nuclear energy. Fossil fuels account for more than 70% of the total energy share worldwide [7].

However, its disadvantages make it an undesirable energy source based on fossil fuels related to climate change and air pollution due to its toxic emissions, which have harmful effects on aquatic life and the surrounding environment. Therefore, the search for renewable energy resources has become a major concern for ensuring energy sources for the coming decades and reducing dependence on fossil fuels to reduce the negative impact on the surrounding environment [6,8].

Many methods are utilized to increase energy generation as by-products. Instead of fossil fuels, thermochemical and biological processes are considered the most attractive technologies for producing clean and inexpensive energy [9,10]. The biological process depends on the activity of microorganisms to produce biogas, which can be used as an important source of heat and energy. One of the most effective biological processes used is anaerobic digestion (AD) [11]. In anaerobic digestion, the microorganisms found in wastewater break down the sludge produced from the primary sedimentation and final settling tanks to produce biogas to produce heat and power. Therefore, anaerobic digestion of the sludge produced is the key solution to its participation in energy production and low operating and maintenance costs compared to the aerobic process, which consumes a large amount of energy to provide oxygen [12]. Therefore, biogas as an energy source can offer several advantages to the economy and the environment compared to other types of fuel produced by the anaerobic digestion process, which works in the absence of oxygen that provides energy costs [11,13].

Many technologies such as activated sludge process (ASP), batch sequencing reactors (SBR), membrane bioreactors (MBR), and aerated lagoons are used for biological treatment. However, membrane bioreactors (MBRs) have appeared to be a promising technology for the biological treatment process. This technology combines the conventional treatment process and filtration using a fiber membrane to settle non-organic matter produced by biological treatment [14]. MBRs can provide several advantages compared to other methods, such as their high efficiency in removing pollutants, as they can achieve a removal efficiency of up to 90% of water pollutants. Furthermore, it is a good choice to upgrade any treatment plant in addition to its small footprint at different stages of the life cycle [15], especially at the multilevel level and in the environmental context, especially at the levels of energy, water, and carbon emissions [16,17].

Therefore, this study is an effective approach to achieving sustainability goals, aiming to address the shortage of natural resources such as energy and water with sustainable alternatives. Furthermore, the experimental work of this study aims to treat the wastewater produced from residual communities to use it in landscaping and irrigation of green areas. Additionally, it uses the sludge produced from the biological treatment process by applying anaerobic treatment to reuse it in heating systems instead of natural gas or electricity to save energy as a step towards improving sustainability.

## 2. Literature Review

Among the 17 Sustainable Development Goals (SDGs) of the United Nations, goal number 11 aims to make future cities resilient and sustainable, while goal number 6 aims to ensure all water and sanitation and sustainable management [18–20]. Green technology as a trend played its role in this context as an alternative to traditional approaches to energy and water issues [21].

The Egyptian state recently paid great attention to sustainable cities as part of its efforts to succeed in the Egyptian National Agenda 2030 [22]. Its Vision 2030 is a unified long-term political, economic, and social vision. It was developed in alignment with the Sustainable Development Goals (SDGs) of the United Nations [23]. This study topic has recently become a major trend in most disciplines in global or local contexts [24]. At the local level, it concerns various issues related to the national context and the long-term vision [25,26], especially water and energy issues. Sustainability rating systems have also emerged at the local level, including divisions, strategies, elements, and relative importance, intensively within the Egyptian context [27].

Achieving the goals established in the SDGs and the 2030 National Vision will require an integrated, long-term approach to Egypt's development path and the potential outcomes and trade-offs from different development scenarios. Recently, energy and water issues have taken a significant position in policy choices and its ability to meet the development goals outlined in the UN's SDGs and the Egyptian Vision 2030, setting some motivations for this study to participate in bridging the already existing gap in the use of wastewater and sludge to harness them to promote the sustainability of Egyptian cities within that vision [24].

Globally, residual zones consume about 40% of total energy consumption in various human activities, such as heating, and are expected to increase to 50% by 2050 [28]. Several sources of energy are used to generate electricity and heat to meet society's needs, such as fossil fuels, neutral gas, coal, and nuclear energy. Fossil fuels account for more than 70% of the total energy share worldwide [7]. However, its disadvantages made it an undesirable source of energy. Fossil fuels are directly related to climate change and air pollution due to their harmful effects on aquatic life and the surrounding environment [29]. The search for renewable energy resources has become a priority to secure energy sources in the coming decades and reduce dependence on fossil fuels to reduce negative environmental impacts as a sustainability requirement [30].

Various methods are used to produce a large amount of energy as by-products. However, thermochemical and biological processes are considered the most attractive technologies for providing clean and inexpensive energy instead of fossil fuels [9]. The biological process depends on the activity of microorganisms to produce biogas, which can be used as an important source of heat and energy. One of the most effective biological processes used is anaerobic digestion [11].

In anaerobic digestion, the microorganisms found in wastewater break down the sludge produced from primary sedimentation and final settle tanks to produce biogas, producing heat and power [31]. Therefore, anaerobic digestion of the sludge produced is the key solution to its participation in energy production and low operating and maintenance costs compared to the aerobic process, which consumes a huge amount of energy to provide oxygen. Therefore, biogas as an energy source can provide several advantages for the economy and the environment compared to other fuels, as it is produced by the anaerobic digestion process, which works in the absence of oxygen and provides energy costs [13].

Many technologies such as activated sludge (ASP), batch sequencing reactors (SBRs), membrane bioreactors (MBRs), and aerated lagoons are used for biological treatment. However, membrane bioreactors (MBRs) seem promising for the biological treatment process. This technology combines the conventional treatment process and filtration using a fiber membrane to settle non-organic matter produced by biological treatment [14]. MBRs can provide various advantages compared to other methods, such as their high efficiency in removing pollutants, as they can achieve a removal efficiency of up to 90% of water pollutants. Furthermore, it is a good option to upgrade any treatment plant in addition to its small footprint in multilevel environmental aspects [16,17].

This study aims to promote sustainability by developing an existing residual compound, as shown in Figure 1, suggesting some solutions to promote sustainability in this compound. It can be achieved by reusing wastewater and using sludge produced by a residential area to produce enough biogas for heating systems through pipeline connections to reduce electricity costs [18]. Furthermore, reuse effluent wastewater after proper treatment to irrigate green areas in residential zones [32].

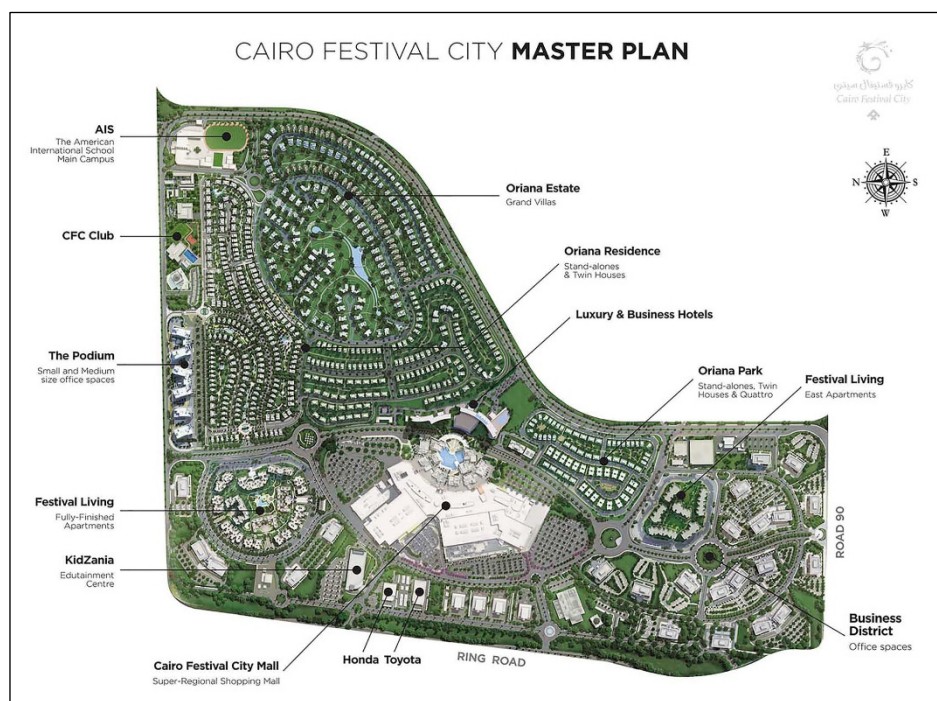

**Figure 1.** Cairo Festival City, New Cairo, Egypt [33].

## 3. Materials and Methods

As shown in Figure 2, which illustrates the conceptual framework of the study, which seeks to promote sustainable cities using sewage and sludge through eight axes, as follows:

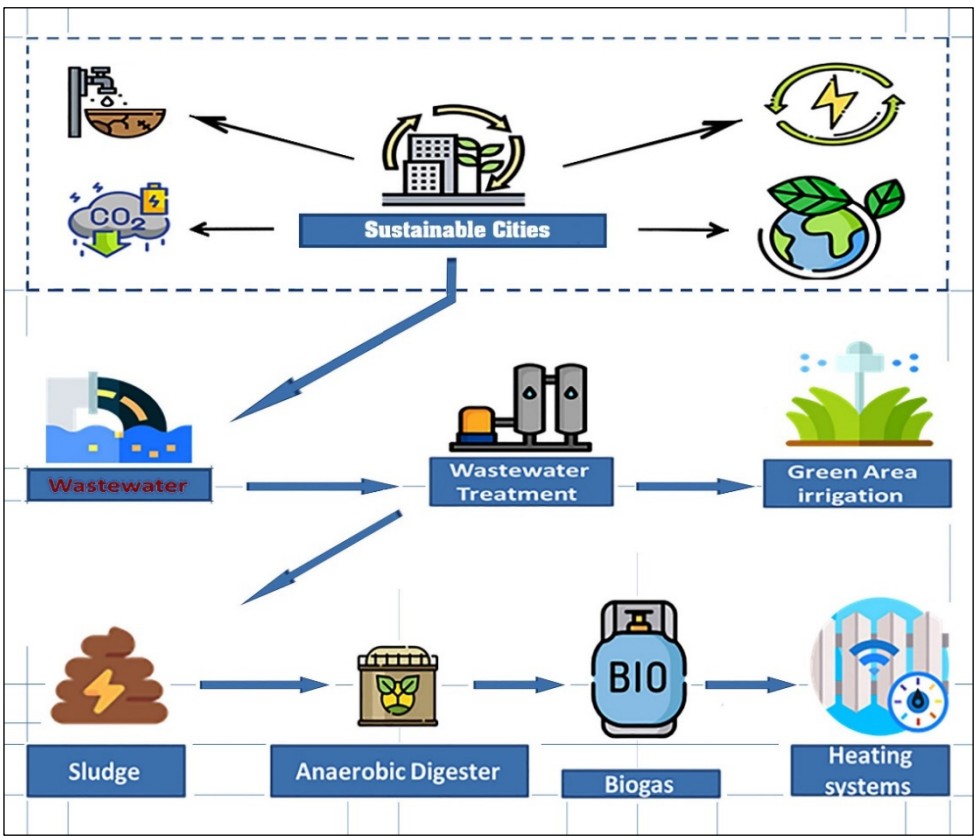

**Figure 2.** Conceptual framework of the study (the authors).

### 3.1. Description of the Existing Residual Community (Compound)

As shown in Figure 1, Cairo Festival City is a compound in Egypt with a direct view of Southern 90th Street, New Cairo City. It establishes new standards as Egypt's premier indoor-outdoor shopping, dining, pedestrian walking, and entertainment destination [34]. It provides amazing residential villas, luxurious apartments, prime office spaces, internationally renowned hotels, international schools, and automotive showrooms within a beautifully landscaped and tranquil community [33].

The compound units are villas, the minimum unit area of 324–1570, the minimum number of bedrooms 4, the minimum number of bathrooms 5, Apartment unit areas range between 151–201 square meters, and bedrooms range between 2–3, and bathrooms range between 2–3. Cairo Festival City is locally considered a creative mixed-use urban community strategically located at the entrance to New Cairo.

### 3.2. Description of Domestic Wastewater

Historically, Egypt has been interested in wastewater treatment programs [35]. In this study, influent wastewater was collected from residual blocks, and this wastewater was a combination of gray and black wastewater collected from each block.

### 3.3. Samples and Sampling

The samples were collected from a maintenance hole inside the residual compound of Egypt. Plastic containers prewashed with dilute water were used to collect wastewater samples. Four samples were taken from the maintenance hole in 10–50 L containers. First, the samples were acidified to fix the BOD and COD values. Then it was moved directly to the National Research Center in Cairo, where the mini-model had been mounted.

### 3.4. Description of the Wastewater Treatment Pilot

Wastewater treatment was performed in a membrane bioreactor plant (MBR). As shown in Figure 3, the pilot consists of a primary sedimentation tank to remove fine matter found in the wastewater. The biological treatment was carried out in an aeration tank in which a column of hollow fiber membrane was submerged. The membrane material was polypropylene with a pore size of 0.1 μm. The aeration tank was equipped with an air blower to provide the required oxygen. The effluent wastewater was passed through the fiber membrane instead of the final settling tank to remove the non-organic containment produced from the aeration tank. Table 1 describes the dimensions of all pilot tanks and the operational parameters of the treatment process. Both primary sedimentation tanks were equipped with scrapers to remove excess sludge from the tank bed.

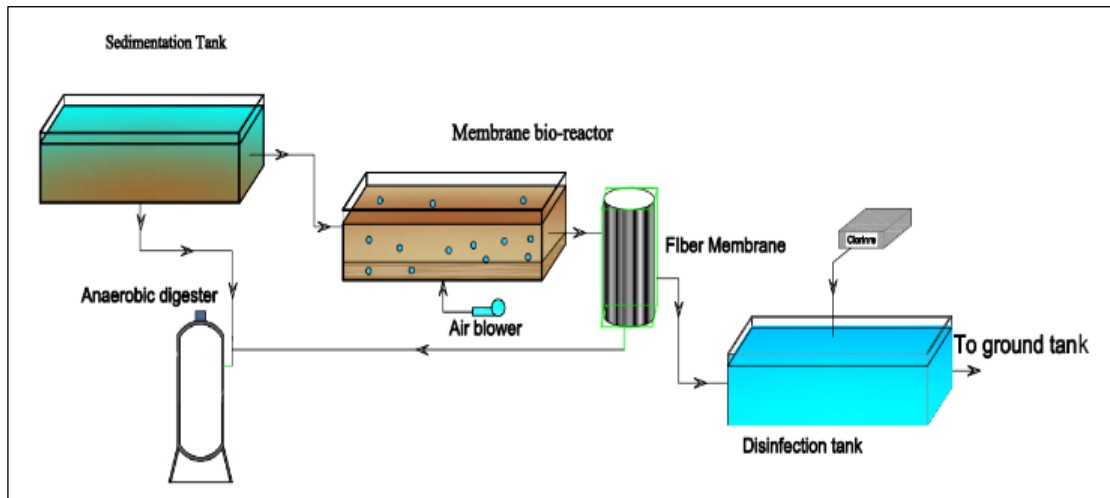

**Figure 3.** Scheme of the wastewater treatment pilot (the authors).

**Table 1.** Description of the wastewater treatment plant.

| Tank | Parameter | Value |
|---|---|---|
| Primary Sedimentation Tank | Water volume | 450 L |
| | Length × width × depth | 1 × 1 × 0.5 m |
| | Hydraulic retention time | 2 and 3 h |
| Aeration tank | Length × width × depth | 1 × 1 × 0.5 m |
| | Water volume | 450 L |
| | Hydraulic retention time | 12 h |
| | Sludge Recirculation | 30% |
| Air blower | Type | Lutz-Jecsco Memdos Smart lp5-Srepper pump |
| | Rate of flow | 5.3 L/h |
| Fiber membrane | Material | Polypropylene |
| | Pore size | 0.1 μm |

### 3.5. Description of the Anaerobic Digester (AD)

The collected sludge from the primary sedimentation and the final clarifier was fed into a cylindrical stainless steel anaerobic digester, as shown in Figure 4; Co-digestion of the produced sludge was carried out to produce biogas. As shown in Table 2, the sludge was delivered for anaerobic digestion through a side piper diameter of 30 cm. The cylindrical digester had a diameter of 50 cm and a height of 20 cm. The outer wall of the tank consists of two layers of stainless steel to keep the temperature constant ($32 \pm 3$ °C). The heat required to digest the sludge was generated using an electric heater. In addition, a mixer with a speed of 100 rpm was placed at the top of the digester. The biogas production rate was recorded daily with a drum-type gas meter.

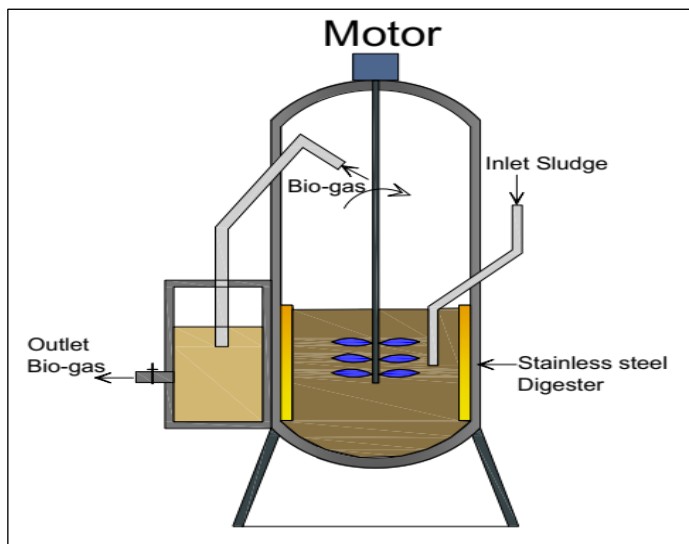

**Figure 4.** Scheme of the anaerobic digester for biogas production (the authors).

**Table 2.** Operating Parameter and the Description of the Anaerobic Digester.

| Digester Unit | Parameter | Value |
|---|---|---|
| Cylindrical digester | Diameter | 50 cm |
| | Height | 20 cm |
| Mixer | Shaft speed | 100 rpm |
| Gas meter | Type | Ritter TG5-model 5 |
| Temperature | Range | $32 \pm 3\,°C$ |
| Sludge | Hydraulic retention time | 20 days |

### 3.6. Experimental Methods

This study established a membrane bioreactor plant to carry out the biological treatment process to reduce pollutant concentrations to a desirable limit. Raw wastewater was collected from a maintenance hole and fed into the primary sedimentation tank. The pilot consists of many units responsible for removing a particular type of pollutant from wastewater, as shown in Figure 3.

All tanks were located at various levels to allow wastewater to pass under gravity. The raw wastewater was fed into a rectangular primary sedimentation tank followed by a biological treatment unit represented in the membrane bioreactor tank with a column of hollow fiber polypropylene membrane with a hydraulic retention time of 12 h. After biological treatment, the wastewater was passed into a rectangular clarifier, where the non-organic substances produced from the biodegradation of the organic matter settled under gravity.

To comply with the limitation of the Egyptian code on reuse, wastewater is used for irrigation purposes. An additional disinfection treatment unit increased treatment efficiency, using chlorine at 30 mg/L. The wastewater from the treated effluent was delivered to the ground tank. The treated wastewater was fed into a sprinkler irrigation network to irrigate the ground area of the compound. The excess sludge from the primary sedimentation tank and the final clarifier was collected and fed into an anaerobic digester tank, where microorganisms break down the sludge to produce biogas at a temperature (35–39 °C) with a hydraulic retention time of 20 days.

### 3.7. Preventing Corrosion in the Biogas Production Process

To avoid the corrupted effects of using raw biogas in combustion equipment, such as corrosion and undesirable emissions. Therefore, a biogas treatment was performed to remove any continents of hydrogen sulfide and the concentration of siloxanes, which is the main reason for pipe corrosion. Therefore, the raw biogas produced by anaerobic digestion passes through a unit filled with activated carbon [12]. The activated carbon was suitable to absorb the concentrations of hydrogen sulfide and siloxanes by converting raw biogas into biomethane, which can be burned in any combustion equipment to generate thermal or electrical energy; The treatment steps were conducted as follows:

As shown in Figure 5, the biogas produced passes through a 5 mm diameter stainless steel pipe to a container of a fixed bed absorbent of activated carbon.

The thickness of the activated carbon absorbent was 20 cm. The container was followed by a flame ionization detector (FID) to measure the biogas rate to adjust the concentration of siloxanes in the stream.

### 3.8. Chemical and Physical Parameters

According to standard methods for examining water and wastewater, BOD, COD, TKN, and TP concentrations were measured at the National Research Center in Cairo.

The temperature was measured daily before taking wastewater samples from the treatment stages.

The determination of the alum dose was measured using a jar test at the National Research Center in Cairo.

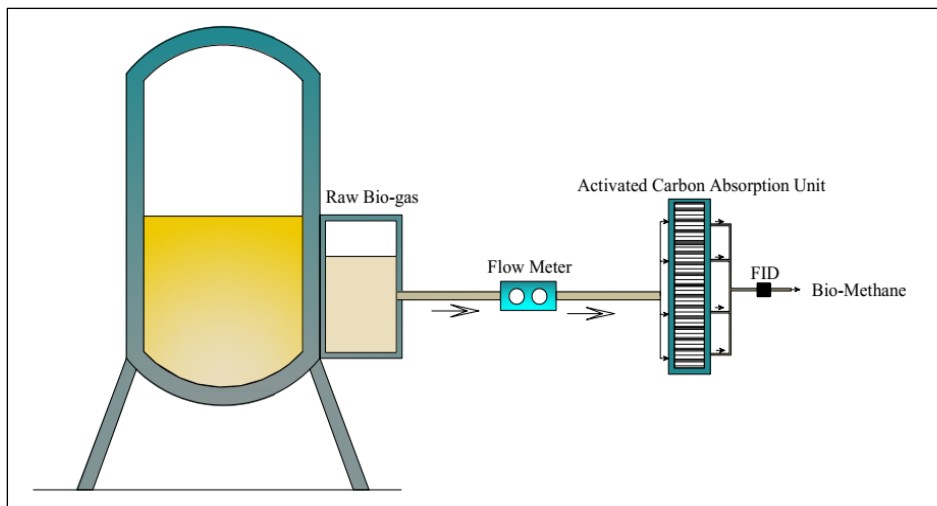

**Figure 5.** Treatment of raw biogas through an activated carbon absorption unit (the authors).

## 4. Results and Discussion

### 4.1. Characteristics of Raw Municipal Wastewater

The physicochemical characteristics of raw wastewater, such as BOD, COD, TSS, TKN, and TP, are summarized in Table 3.

**Table 3.** Physicochemical Characteristics of Raw Domestic Wastewater.

| Parameters | Unit | Samples | | | | Average | Standard Deviation |
|---|---|---|---|---|---|---|---|
| | | 1st | 2nd | 3rd | 4th | | |
| COD | mg/L | 321 | 326 | 346 | 350 | 335.75 | 14.38 |
| BOD | mg/L | 260 | 254 | 267 | 249 | 257.50 | 7.77 |
| pH | - | 8.1 | 8.2 | 7.9 | 7.7 | 7.98 | 0.22 |
| TSS | mg/L | 126 | 120 | 122 | 125 | 123.25 | 2.75 |
| $NH_3$-N | mg/L | 33.4 | 31 | 32.3 | 45 | 35.43 | 6.46 |
| TKN | mg/L | 45 | 52 | 46.5 | 67 | 52.63 | 10.04 |
| TP | mg/L | 6.9 | 7 | 6.7 | 7.2 | 6.95 | 0.21 |

The physicochemical characteristics of domestic wastewater are illustrated in Table 3. Previous pollutants concentrations observed that municipal wastewater was high-strength polluted wastewater. A major environmental threat was loaded with high concentrations of organic matter and nutrients and suspended substances obtained from kitchens, showers, rain, and toilets.

The previous results were the same as those reported by other researchers. Other studies said that the pH of wastewater ranges from 7 to 8.5 [36] and that the pH adjustment does not need to be adjusted to suit the biological treatment process. The mean concentration of COD, BOD, and TSS was 260 mg/L, 145 mg/L, and 95 mg/L, respectively [37]. In Table 4, TN and TP concentrations were 52.63 and 6.95 mg/L higher than the values reported by others.

**Table 4.** Physicochemical properties of effluent wastewater.

| | Treatment Stages | COD | BOD | PH | TSS | NH3-N | TN | TP |
|---|---|---|---|---|---|---|---|---|
| | | All Parameters in mg/L | | | | | | |
| 12 h cycle | Raw sample | 335.75 | 257.5 | 7.98 | 123.25 | 35.43 | 52.63 | 6.95 |
| | After P.S.T for 2 h | 220 | 180 | 7.7 | 61.3 | 29 | 43.2 | 6.1 |
| | After P.S.T for 3 h | 198.1 | 153 | 7.7 | 50.5 | 24.4 | 42.1 | 5.6 |
| | After aeration (12 h) | 36 | 22.6 | 7.9 | 24.6 | 18.3 | 22.1 | 4.7 |
| | After MBRs unit | 26.8 | 18.6 | 7.9 | 4.9 | 3.8 | 5.7 | 1.3 |
| | After disinfection | 23.5 | 14.4 | 7.6 | 3.8 | 3.1 | 4.4 | 1.1 |
| | Removal efficiency (%) | 93 | 94.4 | — | 96.9 | 91.4 | 91.6 | 84.1 |
| Law 48 of the year 1982 | | 100 | 60 | 6–9 | 60 | 40 | — | 10 |

Previous studies indicated that the average concentrations of TN and TP in municipal wastewater were 37.4 and 2.7 mg/L [38]; When comparing the final concentrations of pollutants and the concentrations of raw wastewater from the plant, it was clear that it was higher than the average total concentrations of domestic wastewater. This was due to the mix of gray and black wastewater delivered from the residual area, characterized by high concentrations of pollutants. Furthermore, the large area served by the plant and the high population density led to a higher concentration of wastewater.

### 4.2. Performance of the Membrane Bioreactor Process

As illustrated in Table 4, the effect of each unit of the municipal wastewater treatment plant on the concentrations of COD, BOD, TSS, TN, and TP.

#### 4.2.1. COD Removal

By applying the membrane bioreactor process, influent COD concentrations were 335.75 mg/L and reduced to 23.5 mg/L with a removal efficiency of 93%. This COD removal efficiency was achieved because of the fiber membrane that could abstract the passage of organic matter and macromolecular COD substances due to its tiny pores of 0.1 μm. This COD removal efficiency was lower than that obtained by the others. Previous studies showed that the application of the MBR process was able to achieve a COD removal efficiency of up to 95% [39]. This change in COD removal efficiency may be due to several explanations. This reduction in COD removal efficiency was due to a large amount of organic matter found in domestic wastewater that microorganisms could not biodegrade under aerobic conditions. Furthermore, it may be due to the insufficient hydraulic retention time required to remove all COD concentrations since this plant performed a 12-h aeration cycle and other studies reported that a high COD removal efficiency was achieved with an HRT of 12 to 380 h to achieve a COD removal efficiency of 99% [36,40].

#### 4.2.2. TSS Removal

The MBR technology effectively reduced total suspended solid concentrations from 123.25 mg/L to 3.8 mg/L, with a removal efficiency of 96.9%. This high TSS removal performance was achieved because of the high mechanism of the fiber membrane, which performed a microfiltration process due to its pore sizes, leading to a reduction in efflu-

ent TSS. However, the previous results were slightly lower than those reported in other studies. These studies said that MBR technology achieved a TSS removal of up to 99% [41], which may be due to insufficient bacterial activity that causes a reduction in dissolved oxygen rates.

### 4.2.3. Nitrogen Removal

In Table 4, nitrification and denitrification were carried out in the same tank, and MBR technology was able to reduce TN concentration from 52.63 mg/L to 4.4 mg/L with a removal efficiency of 91.6% and reduce the concentration of NH3-N concentration from 35.43 mg/L to 3.1 mg/L with a removal efficiency of 91.4%. Previous studies showed that the MBR process achieved a TN removal of more than 92.7% [38]. Therefore, the experimental results showed a low performance of MBR under experimental conditions since it achieved a lower removal efficiency than previous studies due to the performance of more than one process in the same basin, the performance of a multifunctional process in the same basin, leading to a limited carbon source and insufficient dissolved oxygen required for the nitrification process [42]. Furthermore, the restricted anoxic areas in the tank needed to perform the denitrification cause a decrease in the denitrification rate, leading to a low TN removal efficiency.

### 4.2.4. Phosphorous Removal

To effectively remove TP, anaerobic zones must be provided to allow polyphosphate-accumulating organisms (PAOs) to release phosphorus. Through this experimental process, MBR reduced the influent TP concentration from 6.95 mg/L to 1.1 mg/L with a removal efficiency of 84.1%. However, other studies illustrated that MBR technology achieved a TP removal efficiency of up to 87%, higher than that achieved in this experiment [38]. This fluctuation in the efficiency of TP removal was due to the limited anaerobic zones that perform all the treatment functions in the same basin, leading to a notable reduction in phosphate release. Furthermore, competition between nitrifying bacteria and organisms that accumulate polyphosphate at the carbon source and dissolved oxygen affected phosphorus uptake.

### 4.2.5. The Effect of Settlement Time on Wastewater Concentrations

Table 4 summarizes the effect of increasing the settling time from 2 to 3 h to achieve high efficiency in removing pollutants from wastewater. Determine the impact of sedimentation time on TSS, COD, BOD, TKN, and TP concentrations. Two separate times (2 h and 3 h) were used. It was clear that increasing the required sedimentation time from 2 to 3 h was more effective in decreasing the physicochemical parameters of the wastewater. It achieved an effective removal efficiency of COD, BOD, TSS, TN, and TP of 40.9%, 40.6%, 59.1%, 20%, and 19.4%, respectively. Furthermore, it is known that the longer the settling time, the higher the number of stable suspended solids under gravity. The previous results were the same as those reported in other studies [43].

### 4.3. *Performance of the Anaerobic Digester for Biogas Production*

The excess sludge was collected from the primary sedimentation tank, and the final clarifier was fed into the aerobic digester. The total sludge produced from the treatment process was approximately 0.32 kg of sludge/day, slightly less than the average sludge produced per day. As reported by [44], other studies estimate that the average amount of sludge produced daily was 0.04 kg/capita/day. In Table 5, the biogas production rate increased over time due to the inverse relationship between the biogas rate and the COD concentration, increasing the HRT. COD decreased, leading to increased biogas production [45].

**Table 5.** The rate of biogas production from the anaerobic digester.

| Time (Day) | 2 | 4 | 6 | 8 | 10 | 12 | 14 | 16 | 18 | 20 |
|---|---|---|---|---|---|---|---|---|---|---|
| Biogas rate (mL/day) | 28.07 | 24.21 | 21.86 | 22.93 | 45.00 | 51.86 | 53.14 | 53.81 | 54.21 | 60.00 |

The low amount of sludge produced by membrane bioreactor technology was due to the low shear load on the sludge produced and the fluctuation in the activity of bacteria, leading to bio-varying biocoenosis [46]. The results are shown in Table 5 on the rate per day of biogas production from the anaerobic digester, which effectively broke down the organic substances loaded within the sludge under a temperature (35 °C) and a hydraulic retention time of 20 days.

To expand the range of biogases produced to cover all residual compound needs, the average biogas produced by the experimental pilot was approximately 41.5 mL/day for a discharge of 500 L/day. Therefore, the total discharge of the compound consisting of 105 blocks is predicted to be 1050 m$^3$/day. By analogy to me, the average biogas rate, the total amount of biogas expected to be obtained from the anaerobic digester, was 38 m$^3$/day. This is enough to provide the installed water heaters in each block with the power needed to meet the occupants' needs.

*4.4. The Imagination of a Full-Scale Domestic Wastewater Treatment Plant*

We applied membrane bioreactor technology as a fully independent compound to build a full-scale wastewater treatment plant model. This wastewater plant must be a compact unit to provide a satisfactory amount of biogas as a source of heat and treated water for irrigation of the green areas of the residual compound. The study illustrates in Table 6 the dimensions and operational parameters of the full-scale wastewater treatment of each unit to achieve concentrations of COD, BOD, TN, and TP to be lower than the limitation of agricultural purposes in the Egyptian code.

**Table 6.** Dimensions and operational parameters of full-scale wastewater treatment.

| Tank | Parameter | Value |
|---|---|---|
| Primary Sedimentation Tank | Tank volume | 45 m$^3$ |
| | Hydraulic retention time | 2 h |
| Aeration tank | Tank volume | 290 m$^3$ |
| | Number of tanks | 2 |
| | Hydraulic retention time | 12 h |
| | Sludge Recirculation | 30% |
| Air blower | Type | Lutz-Jecsco Memdos Smart lp5-Srepper pump |
| | Rate of flow | 50 L/h |
| Fiber membrane | Material | Polypropylene |
| | Pore size | 0.1 μm |
| Anaerobic digester | Diameter | 60 cm |
| | Height | 50 cm |
| | Shaft speed | 100 rpm |
| | Hydraulic retention time | 20 days |

The first unit is expected to be a fine screen with a distance between each bar of 1 cm and inclined by an angle of 60 degrees to the ground to block and remove any floated substance, followed by a rectangular tank to remove colloidal and suspended substances. The biological treatment will be carried out in a membrane bioreactor with a hollow fiber polypropylene membrane column. The plant will be equipped with a chlorine disinfection

unit to increase treatment efficiency. The excess sludge from the plant will be transferred to an anaerobic digester with a diameter of 60 cm and a height of 50 cm.

### 4.5. Suggestions for the Planning of the Irrigation Network

The treated wastewater produced by the pilot was delivered to a ground tank to be reused to irrigate the green areas of the residual compound. From Table 4, the final concentration of COD, BOD, TSS, TN and TP were 23.4 mg/L, 14.4 mg/L, 3.8 mg/L, 4.4 mg/L and 1.1 mg/L. The previous results were lower than the limited COD, BOD, TSS, TN, and TP stated by the Egyptian code for reuse for agricultural purposes [47] to provide enough water to irrigate the entire green area, approximately 200 hectares.

The required amount of water is expected to be greater than 5000 $m^3$/day, and the maximum amount of treated water obtained from the wastewater treatment plant will not exceed 1000 $m^3$/day. Therefore, relying on treated wastewater as the main source of irrigation will not be a good decision due to the insufficiently treated water required for irrigation. Therefore, treated wastewater will be an additional water source to reduce the amount of freshwater consumed for irrigation [48].

### 4.6. Dimensions of Planning for Sustainable City Development

This residual compound met all the conditions and limitations required to be an environmentally friendly city. First, the dependence on treated wastewater for irrigation of the green area was a form of dependence on new and natural resources and the reduction of all amounts of freshwater necessary for other essential fields such as drinking and food preparation [6]. Furthermore, wastewater treatment and safe effluent disposal applied to this residual compound increase the livability of human health. Furthermore, they reduced the hazardous effect on the surrounding environment.

Furthermore, another application of sustainable cities was to reduce the population per capita, since there are less than 50 per capita per hectare, which is consistent with their limitations. Additionally, the reuse of sludge produced from the biological treatment of wastewater from effluents decreased waste output; it is also considered a new renewable energy source and reduces energy production costs.

## 5. Conclusions

According to the 2030 Egyptian Vision, which adopts and aims at sustainable cities, this study proposed a method to develop an existing residual compound in Egypt to integrate the wastewater produced and the surrounding environmental systems. Therefore, these wastes, which are sewage and sludge, should achieve multi-benefits as follows:

The first is to reuse treated wastewater (sewage) in the development and expansion of suitable planned surrounding green areas for irrigation purposes of the softscaping for each residual block. The study proposed MBRs technology by treating an anaerobic sludge digester to provide an additional water source for irrigation.

The second goal is to generate enough biogas for heating systems, and the results showed that using a full-scale wastewater treatment plant would generate enough biogas up to 38 $m^3$/day to cover a major sector with a neutral gas.

This study paved the way for future research on sustainable urban development, particularly in infrastructure related to softscape and energy, with its potential to utilize wastewater in future cities and achieve the Sustainable Development Goals of the United Nations and related targets on a larger scale. Furthermore, to support the Egyptian Vision 2030 goals of sustainable cities locally.

**Author Contributions:** Conceptualization, U.K., W.B., M.N.A. and A.S.E.S.; Data curation, U.K. and M.N.A.; Formal analysis, M.N.A.; Investigation, M.N.A.; Methodology, U.K., W.B., M.N.A. and A.S.E.S.; Resources, U.K. and M.N.A.; Software, M.N.A.; Supervision, U.K. and M.N.A.; Validation, M.N.A.; Writing—original draft, U.K., W.B., M.N.A. and A.S.E.S.; Writing—review & editing, U.K. All authors have read and agreed to the published version of the manuscript.

**Funding:** This research received no external funding.

**Data Availability Statement:** The data files collected can be obtained from the third author on request.

**Acknowledgments:** The authors thank the National Research Center in Cairo for allowing them to use its laboratory.

**Conflicts of Interest:** The authors declare that they have no conflict of interest.

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
