# Peer review of "Sustainability of Egyptian Cities through Utilizing Sewage and Sludge in Softscaping and Biogas Production"

_sustainability, doi:10.3390/su14116675_

Round 1

Reviewer 1 Report

The paper focuses on a compound to propose a model to increase the sustainability of Egyptian cities. To this end, different technologies, such as biological treatment processes, activated sludge, trickling filters, and fixed bioreactors, are used. It also established a pilot laboratory to simulate the sewage treatment plant. Overall, the manuscript is interesting, however, the following issues should be addressed before accepting the article.

  1. Please use a uniform style of writing. For example, see the second paragraph of the introduction on page 1.
  2. In addition, the same citation must not be used in succession. For example, see the second paragraph of the introduction on page 1.
  3. Please mention the novelty and contribution of your work at the end of section 1.
  4. In addition, the manuscript structure must be mentioned at the end of section 1.
  5. The sections, subsections, and subsubsections must be numbered correctly. All of them are 1.
  6. The author should read the paper before submission. For example, on page 4, it is written big and bold, "Error! Reference source not found."
  7. Figure 2 must be placed in the center of the page.
  8. How are the different sample sizes selected in Table 1?
  9. There are many grammatical and formatting errors in the article. The author should remove them.
  10. There must be a discussion section to discuss the study's main results.
  11. The conclusion should be shortened and precise.
  12. The references are not in a uniform style. Some of the information is missing for some references.

Author Response

Dear valued reviewer, it is our pleasure to receive your appreciated comments and constructive criticism. Therefore, we happily detail these comments separately in the following table. Moreover, we are sure it will improve after modifying it according to your appreciated efforts.

During this review, some additional modifications have been made to respond to the three reports received. Additionally, to draw all reviewers' attention to them, we used the "track changes" feature in MS Word, according to the received instructions. Therefore, we did our best to accommodate all the comments received within the maximum compromise.

Finally, we thank you very much for your time and effort, which we greatly appreciate.

Reviewer 2 Report

The manuscript deals with experimental activities and an overview of Sustainability of Egyptian Cities through Utilizing Sewage and Sludge in Softscapes and Biogas Production. The topic is well aligned with the scope of Energies journal. The authors should improve the readability and scientific soundness, the manuscript cannot be accepted in this present form, please carefully revise it to improve the quality, the reviewer has been highlighted several open points below:

  1. The abstract szhould be rewritten to better correspond with merit of article. What problem did you study and why is it important? What methods did you use? What were your main results? And what conclusions can you draw from your results? Please make your abstract with more specific and quantitative results while it suits broader audiences.
  2. Highlihght the novelty aspect
  3. Consder recently published paper in contrast to your work: https://doi.org/10.1007/s10098-021-02103-1 
  4. Add more details on biogas you reported.
  5. Add the statistical data of obtained results
  6. PLZ, arrage properly sub-divisions of your manuscript
  7. Add the proper chart, that makes your results more visible and understranding.
  8. Extenfd discussion!

Author Response

(The authors gave the same response as above.)

Reviewer 3 Report

The manuscript requires rigorous revision as suggested by the reviewer in the attached file.

Author Response

(The authors gave the same response as above.)

Round 2

Reviewer 1 Report

As the authors have addressed my concerns.

Author Response

Dear valued reviewer,

It is our pleasure to receive your appreciated minor comments for this second round. Therefore, we are appreciating all your efforts.

Reviewer 2 Report

The Authors revised manuscript accordingly, in its present form can be accepted for publication.

Author Response

(The authors gave the same response as above.)

Reviewer 3 Report

The authors have considerably improved the manuscript after incorporating the review comments and suggestions. I appreciate the same.

Author Response

(The authors gave the same response as above.)
